# Isolation and Genetic Characterization of a Bovine Coronavirus KBR-1 Strain from Calf Feces in South Korea

**DOI:** 10.3390/v14112376

**Published:** 2022-10-27

**Authors:** Jihye Shin, SeEun Choe, Gyu-Nam Park, Sok Song, Ki-Sun Kim, Byung-Hyun An, Bang-Hun Hyun, Dong-Jun An

**Affiliations:** 1Virus Disease Division, Animal and Plant Quarantine Agency, Gimcheon 39660, Korea; 2Department of Veterinary Medicine Virology Laboratory, College of Veterinary Medicine and Research Institute for Veterinary Science, Seoul National University, GwanAk-Ro 1, GwanAk-Gu, Seoul 08826, Korea

**Keywords:** BCoV, goat, calf, antibody, adjuvant, phylogenetic tree

## Abstract

Bovine coronavirus (BCoV) causes severe diarrhea in neonatal calves, winter dysentery in adult cattle, and respiratory disease in feedlot cattle, resulting in economic losses. A total of 16/140 calf diarrheic feces samples collected in South Korea between 2017 and 2018 were positive for BCoV. Phylogenetic analysis of the complete spike and hemagglutinin/esterase genes revealed that the 16 Korean BCoV strains belonged to group GIIa along with Korean strains isolated after 2000, whereas Korean BCoV strains isolated before 2000 belonged to group GI. Mice and goats inoculated with an inactivated KBR-1 strain (isolated from this study) generated higher antibody titers (96 ± 13.49 and 73 ± 13.49, respectively) when mixed with the Montanide01 adjuvant than when mixed with the Carbopol or IMS1313 adjuvants. Viral antigens were detected in the large intestine, jejunum, and ileum of calves inoculated with inactivated KBR-1 vaccine (10^4.0^ TCID_50_/mL) at 14 days of post-challenge (DPC). However, no viral antigens were detected in calves vaccinated with a higher dose of inactivated KBR-1 strain (10^6.0^ TCID_50_/mL) at 14 DPC, and they had high antibody titers and stable diarrhea scores. Currently, the group GIIa is prevalent in cows in South Korea, and although further research is needed in the future, the recently isolated KBR-1 strain has potential value as a new vaccine candidate.

## 1. Introduction

Bovine coronavirus (BCoV) is a member of the species *Betacoronavirus* within the family *Coronaviridae* [1]. BCoV possesses a single-stranded, positive-sense, non-segmented RNA genome of approximately 31 kb, which encodes five major structural proteins: nucleocapsid (N), transmembrane (M), spike (S), small membrane (E), and hemagglutinin/esterase (HE) [2]. Of these, the S protein is a particularly important mediator of host cell attachment, viral entry, and pathogenesis [3]. The S protein comprises the S1 subunit, which contains the dominant neutralizing epitopes, and the S2 subunit, which mediates viral membrane fusion. The HE protein exhibits an esterase receptor-destroying activity that may also be important for viral entry and serves as a second viral attachment protein to initiate infection [4].

BCoV is shed in the feces and nasal secretions, and is associated with three distinct clinical syndromes in cattle [5]: (neonatal) calf diarrhea [(N)CD)] [6,7]; winter dysentery (WD), characterized by hemorrhagic diarrhea in adults [7,8,9,10]; and respiratory infections in cattle of different ages (commonly observed as a part of the bovine respiratory disease complex, or shipping fever in feedlot cattle) [6,11,12]. In South Korea, WD and CD outbreaks were confirmed by surveys conducted during 2002–2004 [13] and 2004–2005 [14], respectively. BCoV isolated from Korean WD cases suggests that the virus has dual tropism and induces pathological changes in both the digestive and respiratory tracts of calves [14]. Another study showed that 1% of Korean goats have antibodies against BCoV [15], whereas another detected a BCoV-like virus in nasal swabs from non-captive Korean wild water deer [16]. A recent study performed genetic characterization of BCoV isolated from diarrheic pre-weaned native Korean calves from 2019–2021 [17]. Because BCoV is common in Korean cows that have recently been inoculated with the BCoV vaccine, cow farm owners and clinical veterinarians are proposing development of a new vaccine.

Here, we compared the sequences of the S and HE genes of BCoV strains detected in this study and compared them with BCoV strains isolate around the world. A vaccine candidate strain KBR-1 was used to inoculate goats and mice, and immunogenicity and optimal adjuvant efficacy were investigated. In addition, we also investigated the potential of the KBR-1 strain as a vaccine candidate for calves to protect them against infection by BCoV.

## 2. Materials and Methods

### 2.1. Diarrhea Samples and RNA Extraction

A total of 140 diarrhea samples were obtained from calves under 2 months-of-age. Samples were collected from farms by three clinical veterinarians between February 2017 and May 2018. Samples were collected from eight provinces: nine samples (three farms) from Gyenggi (GG), 11 samples (five farms) from Gangwon (GW), 38 samples (13 farms) from Chungnam (CN), six samples (three farms) from Chungbuk (CB), 24 samples (nine farms) from Gyeongnam (GN), eight samples (five farms) from Gyeongbuk (GB), nine samples (four farms) from Jeonbuk (JB), and 35 samples (14 farms) from Jeonnam (JN). A total of 57 and 83 calves were either vaccinated or unvaccinated against BCoV, respectively. Total RNA was extracted from 40–60 mg of fecal sample using a Qiagen RNeasy kit, and cDNA was prepared using a Helix Cript^TM^ Easy cDNA synthesis kit.

### 2.2. RT-PCR and Sequences Analysis of BCoV

Total RNA extracted from fecal samples was subjected to a nested RT-PCR assay targeting the BCoV nucleocapsid gene; the primers and methods have been described previously [18]. Samples identified as BCoV-positive by the nested RT-PCR were subjected to complete gene amplification using six primer pairs targeting the S gene, and two primer pairs targeting the HE gene, as described previously [18]. The amplified PCR products were extracted and subcloned into the pGEM-T vector system II (Promega, Madison, WI, USA). Cloned plasmids corresponding to the BCoV S and HE genes were sequenced with T7 and SP6 sequencing primers and an ABI Prism 3730xl DNA sequencer (Cosmo Genetech Co., Seoul, Korea). After the genes from each strain were sequenced, multiple sequence alignment of the complete S and HE genes, as well as phylogenetic analysis, were carried out using Bio Edit Sequence Alignment Editor version 7.2 and MEGA 7.0 software [19], respectively. Bootstrap values based on 1000 replicates were used to assess the confidence limits of the branching. Phylogenetic trees were constructed based on the complete S and HE gene sequences of BCoV reference strains isolated in Asia, America, and Europe, including the 1994–2021 strains detected in South Korea and the OC43 strain (human coronavirus) was used as an outgroup. Complete S and HE gene sequences of BCoV reference strains were obtained from the National Center for Biotechnology Information (NCBI) GenBank database. The accession numbers of the reference sequences reported worldwide including the Korean strains are displayed in Figure 1 and Figure 2.

### 2.3. Isolation of BCoV

Monolayers of human rectal tumor (HRT-18) cells were used to isolate BCoV. The BCoV isolate was exposed to antibiotics from the initial to the 3rd passage, and the absence of major bovine enteric viruses (bovine rotavirus and bovine viral diarrhea virus) was confirmed at every passage using RT-PCR (LiliF-BD-Multi RT-PCR kit, iNtRON Biotechnology, Seongnam, Republic of Korea). In brief, supernatants from fecal samples were collected by centrifugation and passed through 0.45 µm filters. The fecal supernatants were treated for 30 min with 10 μg/mL trypsin (Gibco, Grand Island, NY, USA), with intermittent rocking. The cultured cells were then washed twice with plain medium and inoculated with fecal supernatant for 1 h. After the inoculum was discarded, cells were cultured for 3–4 days at 37 °C/5% CO_2_ in RPMI 1640 (Gibco, Grand Island, NY, USA) containing 5 μg/mL trypsin. The cells were observed daily for evidence of cytopathic effects (CPE); if no CPE were observed, isolation of BCoV was confirmed in an indirect immunofluorescence assay (IFA). IFA was performed with a mouse monoclonal antibody specific for BCoV (Invitrogen, Carlsbad, CA, USA), followed by staining with a fluorescence-conjugated goat anti-mouse antibody (anti-FITC, green; SeraCare, Gaithersburg, MD, USA). Nuclei were stained with DAPI (4′, 6-diamidino-2-phenylindole, blue; Sigma-Aldrich, St. Louis, MO, USA). Merged images (FITC and DAPI) were used for analysis. Virus titers (TCID_50_) were determined using the Reed and Muench method [20]. In brief, BCoV was serially diluted 10-fold in RPMI 1640 medium containing 5 μg/mL trypsin, and then inoculated into HRT-18 cells pre-seeded in 96-well plates (eight replicate wells per dilution). After incubation for 60 min, the medium was replaced with RPMI 1640. The plates were incubated at 37 °C/5% CO_2_ for 3 days. The medium was discarded from the wells and the plates were immunoreacted with an anti-BCoV monoclonal antibody for IFA. Every well containing FITC-stained cells was counted, and titers were expressed as TCID_50_/mL. The KBR-1 strain isolated at passage 40 was used for adjuvant selection in mice and goats, and to evaluate vaccine efficacy in calves.

### 2.4. Adjuvant Selection Using Mice and Goats

The KBR-1 (10^5.0^ TCID_50_/mL) strain was obtained from HRT-18 cells and inactivated using binary ethyleneimine (BEI). The residual BEI in the inactivated viral fluid was neutralized by sodium thiosulfate. Three adjuvants (Carbopol (0.02% *w/v*, Lubrizol, USA) [21]; Montanide01 (10% *v/v*, SEPPIC, France); and IMS1313 (30% *v/v*, SEPPIC, France)) were mixed with the inactivated KBR-1, in accordance with the manufacturer’s instructions (SEPPIC, France). Three groups of 12 goats (10 months-old) and three groups of 15 mice (15–20 g) were inoculated intramuscularly (IM) with three adjuvants to identify the best one. For each adjuvant, three goats/group were inoculated IM (1 mL/dose) twice, with an interval of 3 weeks; in addition, five mice received a dose of 0.2 mL. Three goats (10 months-old) and five mice (15–20 g) were used as non-vaccine mock controls. Blood samples were collected three times (before vaccination, at 3 weeks after the first vaccination, and at 4 weeks after the second vaccination) to detect BCoV antibodies.

### 2.5. Immunization of Calves and Examination of Vaccine Efficacy

All six of the calves used for vaccine efficacy evaluation were negative for BCoV antigen and antibodies. The RT-PCR kit (LiliF-BD-Multi RT-PCR kit, iNtRON Biotechnology, Seongnam, Republic of Korea) for cow enteric viruses (BCoV, bovine rotavirus A, and bovine viral diarrhea virus) was used to screen the antigen-negative calves, and an hemagglutination inhibition (HI) test was performed to screen BCoV antibody-negative calves. Four of six calves (2 months-old) per group were inoculated with a high (10^6.0^ TCID_50_/mL; *n* = 2) or low (10^4.0^ TCID_50_/mL; *n* = 2) dose of KBR-1 vaccine. The remaining two calves comprised the mock group (without vaccine). Calves were vaccinated twice (1 mL per dose), with a 3-week interval in-between. At 2 weeks after the second vaccination, all calves were inoculated orally with 5 mL (10^6.0^ TCID_50_/mL) of virulent BCoV. Virulent BCoV was obtained from KVCC (Korea Veterinary Culture Collection, Republic of Korea, KVCC number: VR1800062). Blood, nasal, and fecal samples were collected at 0, 21, and 35 DPV (days post-vaccination), and at 1, 2, 3, 6, 9, 12, and 14 DPC (days post-challenge) to measure antibody and antigen levels. Fecal consistency was scored as follows: 0, solid; 1, pasty; 2, semi-liquid; and 3, severe liquid. Autopsy was performed at 14 DPC, and organs were tested for BCoV.

### 2.6. Serum Neutralizing Antibody and HI Tests

To measure serum neutralizing antibody titers, mouse and goat sera were diluted serially 2-fold in serum-free RPMI 1640 medium (Gibco) containing trypsin, and then mixed with 10^2.0^ TCID_50_/0.1 mL BCoV. The virus-serum mixture was incubated for 1 h to neutralize the virus and then used to infect HRT-18 cells. After incubation for 3 days, plates containing cells not expressing serum neutralizing BCoV antigen were assessed by IFA. Neutralizing antibody titers were expressed as the reciprocal of the highest dilution that yielded 50% neutralization. Calf antiserum was mixed with a receptor-destroying enzyme to remove non-specific inhibitors. Next, 4–8 HA units of BCoV were added to serially diluted sera. Finally, 1% mouse erythrocytes (RBC) were added to generate RBC-BCoV antigen conjugates. An HI titer ≥ 1:20 was considered positive.

### 2.7. Real-Time RT-PCR

Real-time RT-PCR was used to quantify BCoV antigens from nasal and fecal samples. Total RNA was extracted using a micro column-based RNeasy Mini kit (Qiagen, CA, USA), and the real-time RT-PCR targeting the N gene [22,23] was performed using the Path-IDTM Multiplex One-Step RT-PCR kit (Life Technologies, Carlsbad, CA, USA). BCoV PCR primer sets (F primer: CTA GTA ACC AGG CTG ATG TCA ATA CC; R primer: GGC GGA AAC CTA GTC GGA ATA) and probe (FAM-CGC CTG ACA TTC TCG ATC–MGB) were used for the real-time RT-PCR [22,23]. The cycling conditions of real-time RT-PCR were as follows: reverse transcription (RT) for 30 min at 45 °C; activation of DNA polymerase for 10 min at 95 °C; 40 cycles of denaturation at 94 °C for 15 s, and annealing/extension at 60 °C for 60 s.

### 2.8. Statistical Analysis

All statistical analyses were performed using GraphPad Prism software, version 6.0, for Windows. Data are expressed as the mean ± SD.

## 3. Results

### 3.1. Prevalence of BCoV

The prevalence of BCoV on calf farms reporting diarrhea was 11.4% (16/140). In terms of age, six calves were between 1 and 20 days old, six were between 21 and 30 days old, and four were between 31 and 60 days old. With respect to regions, BCoV was detected in GW (1/11, 9.1%), CN (6/38, 15.8%), GN (2/24, 8.3%), JB (2/9; 22.2%), and JN (5/35, 14.3%). No cases were detected in GG (0/9), CB (0/6), or GB (0/8).

### 3.2. Phylogenetic Analysis of the S and HE Genes

Phylogenetic analysis revealed that the complete S gene was split into two groups (GI and GII). The GI group includes old strains (European, American, and Canadian), including classical strains (Figure 1). GII also split into two subgroups: GIIa (America and Asia) and GIIb (European). The GIIa subgroup contains Korean BCoV strains isolated between 2002 and 2021, including all of the strains detected in this study (Figure 1); it also included strains from Japan, China, Vietnam, Cuba, and the USA (Figure 1). The GIIb subgroup included independent strains from European countries, including France, Sweden, and Denmark (Figure 1). Phylogenetic analysis of the HE gene also revealed a split into two groups. Korean strains (SUN5, A3, and BC94) detected in 1994 were included in the GI subgroup, and all Korean strains detected between 2002 and 2021 were included in the GIIa subgroup (Figure 2). The 16 Korean strains detected in 2017 and 2018 were closely related to strains detected in Korea between 2019 and 2021 (Figure 2).

### 3.3. Comparison of the S and HE Gene Sequences

Comparative analysis of nucleotide (nt) sequence homology revealed that the S and HE genes of the three Korean strains (SUN5, A3, and BC94) detected in 1994 showed high similarity (99.5–99.6% and 99.8%, respectively) with the Mebus strain, which is virulent (Table 1).

However, the nt sequences of the S genes of Korean strains detected in 2017 and 2018 showed somewhat lower similarity (97.3–97.8% and 97.5–97.6%, respectively) with the Mebus strain (Table 1). In addition, the genes (S and HE genes) of strains isolated in Korea after 2000 showed low homology with the Mebus strain (Table 1). Comparison of the deduced amino acid sequences for the S protein of Korean BCoVs showed that the S protein of the Mebus strain has high similarity with that of SNU5 and BC94 strains isolated in 1994, although it harbors mutations in 34 amino acids when compared with Korean strains from 2017–2018 (Table 2). The HE gene of the Mebus strain harbored mutations at six amino acid positions when compared with the Korean BCoV strains from 2002–2021 (Table 3).

### 3.4. Non-CPE of the BC94 and KBR-1 Strains

Of the 16 Korean BCoVs examined herein, only the KBR-1 strain, detected in the feces of a calf in the CN region in 2017, could be passaged in human rectal tissue 18 (HRT-18) cells. The virus yielded 10^3.0^ TCID_50_/mL at passage 5, 10^4.2^ TCID_50_/mL at passage 20, and 10^6.2^ TCID_50_/mL at passage 40 (Figure 3A). The highest titers were detected between 48 (10^6.2^ TCID_50_/mL) and 72 h (10^6.0^ TCID_50_/mL) of culture (Figure 3B). The non-CPE of the KBR-1 strain was confirmed by IFA, staining with DAPI, on examination of merged images (FITC and DAPI), or on bright field microscopy (Figure 3C).

### 3.5. Antibody Titers and Protection of Mice and Goats Inoculated with the Inactivated KBR-1 Vaccine

Anti-BCoV antibody titers in mice at 2 weeks after the second inoculation with the inactivated KBR-1 vaccine were as follows: 2^5.3^ log_2_ when mixed with Carbopol (10% v/v, Lubrizol, USA); 2^6.4^ log_2_ when mixed with Montanide01 (50% *v/v*, SEPPIC, France); and 2^4.8^ log_2_ when mixed with IMS1313 (10% *v/v*, SEPPIC, France) (Figure 4). The BCoV antibody titers in goats at 2 weeks after the second injection of the inactivated KBR-1 vaccine were as follows: 2^6.4^ log_2_ when mixed with Carbopol, and 2^7.1^ log_2_ when mixed with Montanide01, or 2^7.0^ log_2_ when mixed with IMS1313 (Figure 4). After two inoculations with the inactivated KBR-1 vaccine, neither mice nor goats showed specific clinical signs.

### 3.6. Diarrhea Score, and Antigen and Antibody Levels, after Challenge of Calves with the Inactivated KBR-1 Vaccine

Two calves receiving the inactivated KBR-1 (10^6.0^ TCID_50_/dose) vaccine mixed with the Montanide01 adjuvant showed a titer of 40–80 HI units at 2 weeks after the second vaccine, and 160 HI units at DPC 14; no antigen was detected in nasal, fecal, and blood samples during the observation period after vaccination and challenge (Table 4). During the observation period, calves vaccinated with the inactivated KBR-1 (10^6.0^ TCID_50_/dose) had normal feces (Table 4). However, calves vaccinated with a lower dose of the inactivated KBR-1 (10^4.0^ TCID_50_/dose) vaccine had low antibody titers (20 HI units at DPV 35 and 40–80 HI units at DPC 14) (Table 4).

BCoV was detected in feces at DPC 1–3 and at DPC 2–6, respectively, with a diarrhea score of 1 and 2 at DPC 3 and 6, respectively (Table 4). For all calves in the mock group exposed to virulent BCoV, virus was detected in rectal swabs for 8–10 days (at DPC 3–12 and at DPC 2–9), which was longer than in the vaccination groups. The mock group also developed diarrhea, with a diarrhea score of 1 at DPC 2–14 and 2 at DPC 2–12 (Table 4). Real-time RT-PCR showed that all nasal samples were negative. However, in calves vaccinated with a lower dose of the inactivated KBR-1 (10^4.0^ TCID_50_/dose) vaccine, virus was detected in feces for 1–3 DPC (ct value: 28.7–33.1) and 2–6 DPC (ct: 29.1–32.9) post-challenge (Table 4). In addition, virus was detected in feces from the mock group for 3–12 DPC (ct: 20.7–31.1) and 2–9 DPC (ct: 22.5–29.2) (Table 4). None of the calves died during the experimental period. All calves were euthanized and autopsied at DPC 14. No BCoV was detected in 13 organs (including brain) harvested from calves vaccinated with the high dose of inactivated KBR-1 (10^6.0^ TCID_50_/dose) vaccine (Table 5).

However, calves vaccinated with the lower dose of inactivated KBR-1 (10^4.0^ TCID_50_/dose) vaccine harbored virus in the organs, particularly the large intestine (ct: 32.6 and 33.2), jejunum (ct: 28.9 and 30.5), and ileum (ct: 26.1 and 29.8) (Table 5). In addition, the mock group harbored virus in the organs, particularly the duodenum (ct: 31.8), jejunum (ct: 23.6 and 24.7), and ileum (ct: 26.2 and 25.5) (Table 5).

## 4. Discussion

Between 2010 and 2019, many countries reported positive BCoV infections in calf feces [24], including China (12.2–19.0%), Thailand (12%), India (16%), Vietnam (6.9%), Australia (21.6%), Iran (7.2%), and Algeria (20.7%). In Korea, the prevalence of BCoV in 2002–2003 was 33.0% [25] and remained at 5.4–15.6% until 2021 [13,17]. Although the overall prevalence of BCoV in Korea has tended to decrease, continuous and damaging BCoV infections on cow farms might be possible. The present study shows that the prevalence of BCoV in diarrheic calves on cow farms located nationwide is 11.4%. Prevalence rates are particularly high in Jeonbuk (22.2%), Chungnam (15.8%), and Jeonam (14.3%) provinces. This result shows that continuous epidemiological monitoring is required to reduce future BCoV outbreaks.

In this study, phylogenetic tree analysis of the S and HE genes identified three branches: GI (classical strains), GIIa (America/Asia strains), and GIIb (European strains). The phylogenetic tree of S and HE genes shows that Korean strains detected in 1994 belonged to GI, but Korean BCoV strains detected from 2002 to 2021 belonged to GIIa. All HE genes of Korean BCoV strains detected after 2002 were closely related to the R-AH187 strain (isolated in the USA in 2000) and all S genes were closely related to the LSU-94LSS-051-2 strain (isolated in the USA in 1994). Alignment of the putative amino acid sequences derived from the S genes of the Korean BCoV strains identified from 1994, 2002–2018, and 2019–2021 in this study highlighted differences in the dominant amino acids at positions 25, 34, and 35 of the S protein (compared with the Mebus strain), respectively. A previous study suggests that Korean BCoVs show more genetic variation than those isolated in other countries due to continuous evolution [17]. The most recent common ancestor (TMRCA) results published in a recent evolutionary study also suggest that Korean BCoVs have been influenced by viruses originating from the USA, which have adapted rapidly to their environment [17]. The BCoV S and HE gene sequences identified in South Korea have formed independent clusters since 2002; in particular, the 16 BCoVs from 2017–2018 are closely associated with the cluster of 15 BCoVs from 2019–2021. The fact that BCoV strains emerging recently in South Korea are included in a single cluster is thought to be due to continuous evolution within the same cluster since 2002. In addition, it is estimated that evolution within the same cluster will continue if BCoV does not flow from other countries.

Although further investigations are needed, ongoing outbreaks have led cow disease experts to suggest that the efficacy of current vaccines requires improvement. Of the samples used in the present study, 40.7% (57/140) were collected from calves born to 13 vaccinated cow farms and 59.3% (83/140) from calves born to 25 unvaccinated cow farms. However, most of the 16 BCoV-positive diarrhea cases identified in this study were calves born to vaccinated cows (62.5%; 10/16) in four farms, whereas 37.5% (6/16) were in calves born to unvaccinated cows in three farms. Given the high prevalence in BCoV-vaccinated farms, future studies should investigate whether the current BCoV vaccine protects against the most recent virulent BCoV strain.

The BCoV antibody titers in goats at 2 weeks after the second injection of the inactivated KBR-1 vaccine (10^5.0^ TCID_50_/dose) in this study were high (2^7.1^ log_2_) when mixed with Montanide01 and when mixed with IMS1313 (2^7.0^ log_2_); however, it was low (2^6.4^ log_2_) when mixed with Carbopol. IMS1313 and Montanide01 are commercial adjuvants comprising nanoparticles and an immunostimulatory compound, and a synthetic polymer, respectively (SEPPIC). According to the manufacturer, the two adjuvants enhance the strength and duration of the immune response. Montanide01 is an aqueous-based gel adjuvant, which facilitates easier vaccine delivery than IMS1313, which is emulsified in oil. In addition, we selected Montanide01 for the cattle experiments because it is recommended by the manufacturer as a vaccine adjuvant for cattle and small ruminants.

Here, we found that the candidate KBR-1 (10^6.0^ TCID_50_/dose) vaccine mixed with the Montanide01 adjuvant protected calves against diarrhea and virus shedding, with little antigen detected in organs following challenge with a virulent virus strain; however, the inactivated KBR-1 vaccine at a lower dose (10^4.0^ TCID_50_/dose) failed to protect against diarrhea and virus shedding, with viral antigen detectable in organs. The ct values (28.7–33.1 and 29.1–32.9) obtained for real-time RT-PCR of feces samples collected after challenge from calves vaccinated with inactivated KBR-1 (10^4.0^ TCID_50_/dose) were relatively higher than those (20.7–31.1 and 22.2–29.2) from feces collected from mock controls after challenge. In addition, after euthanasia at DPC 14, the ct values were higher for organs in mock group calves than in those from inactivated KBR-1 group calves. These results may imply a weak protective effect, even in calves vaccinated with the inactivated KBR-1 (10^4.0^ TCID_50_/dose) strain. A previous study suggested that an inactivated vaccine may be useful for preventing WD caused by BCoV [26]. Thus, using the candidate KBR-1 vaccine at a high concentration may be necessary for effective protection of newborn calves from CD or WD.

## 5. Conclusions

In conclusion, phylogenetic analysis revealed that all Korean BCoV strains isolated after 2002 belong to the GIIa group. Animal studies showed that the recently isolated KBR-1 vaccine strain (group GIIa) candidate was effective and safe and protects cows from infection by virulent BCoV. Although additional experiments with a large number of animals are required, the KBR-1 strain shows potential as a vaccine candidate.

## Figures and Tables

**Figure 1 viruses-14-02376-f001:**
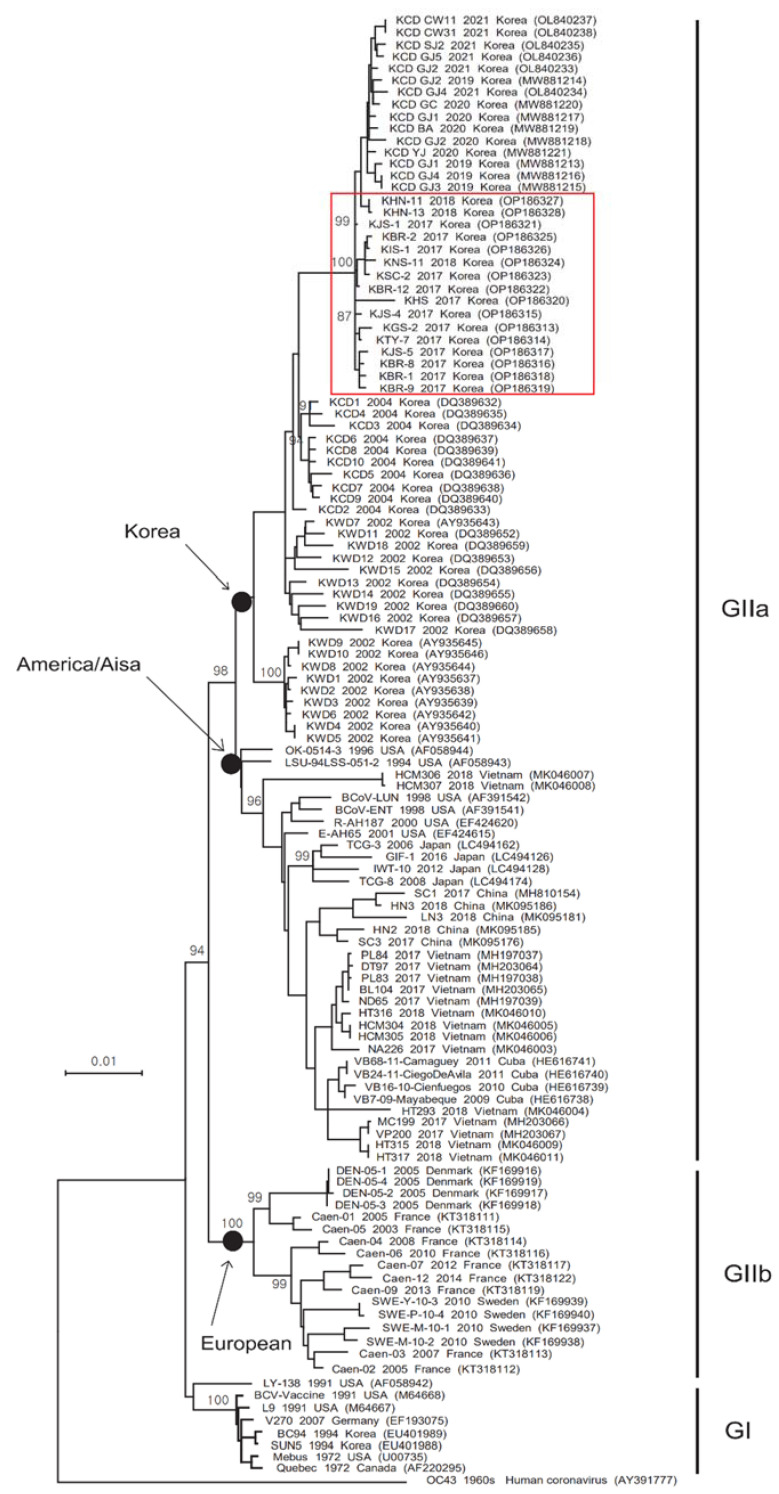
Phylogenetic tree based on the complete S gene sequences of BCoV reference strains isolated worldwide, including the 2017–2018 strains detected in this study. A maximum-likelihood phylogenetic (LogL = −15690.97) tree was constructed using the Tamura-Nei model and the Nearest-Neighbor-Interchange method. Bootstrap values (1000 replicates) ≥ 70% are shown at the nodes, and the sequences identified in this study are marked by a red box.

**Figure 2 viruses-14-02376-f002:**
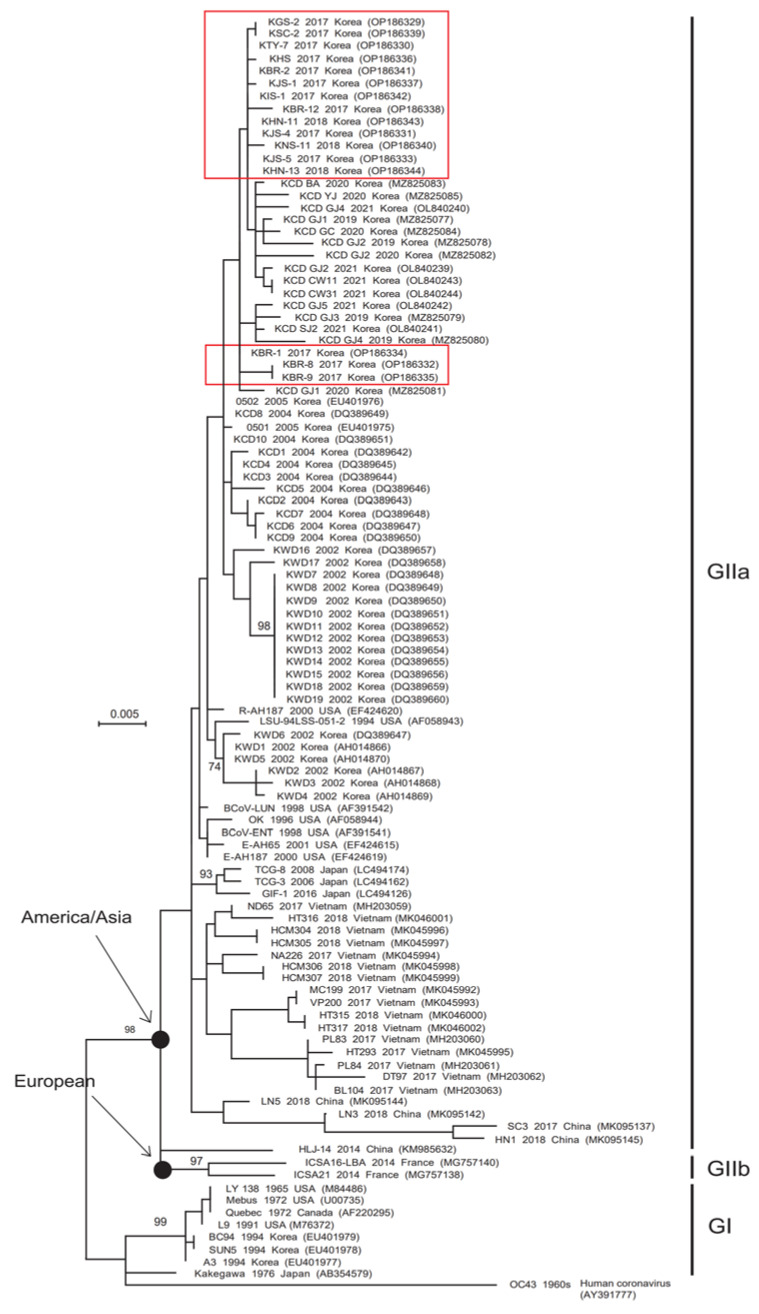
Phylogenetic tree based on the complete HE gene sequences of BCoV reference strains isolated worldwide, including the 2017–2018 strains detected in this study. The phylogenetic (LogL = −3940.11) tree was constructed using the maximum-likelihood method (Tamura-Nei model), with bootstrap values (1000 replicates) ≥ 70%, in MEGA 7.0 software. The sequences identified in this study are marked by a red box.

**Figure 3 viruses-14-02376-f003:**
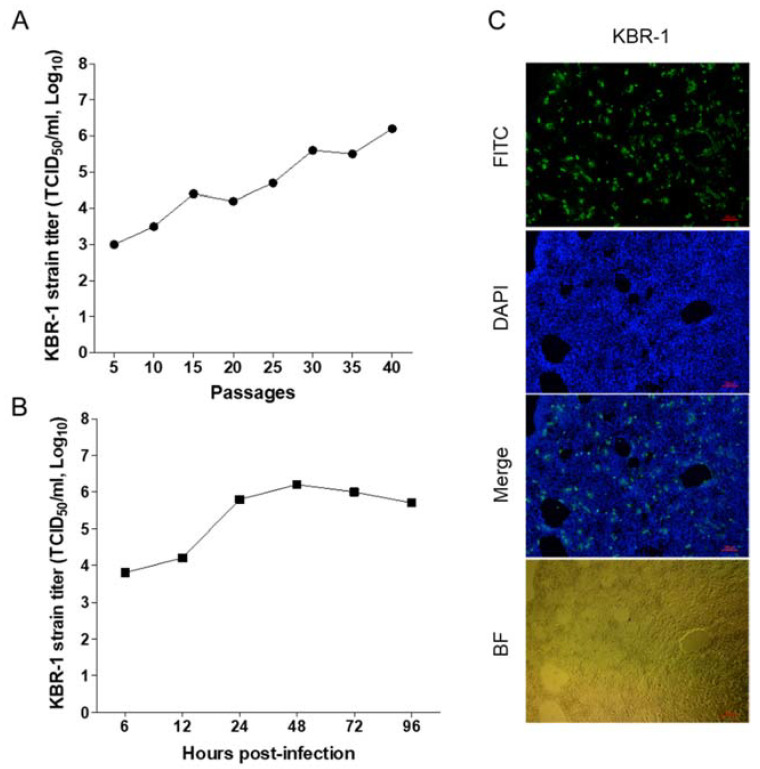
Proliferation and isolation of the KBR-1 strain using HRT-18 cells. Virus titer (according to passage number of the KBR-1 strain) detected in diarrhea samples from calves (**A**); virus titer at different times after inoculation with the KBR-1 strain (**B**); and virus detection and CPE (IFA, DAPI, merged images, and bright field microscopy) of the KBR-1 strain (**C**).

**Figure 4 viruses-14-02376-f004:**
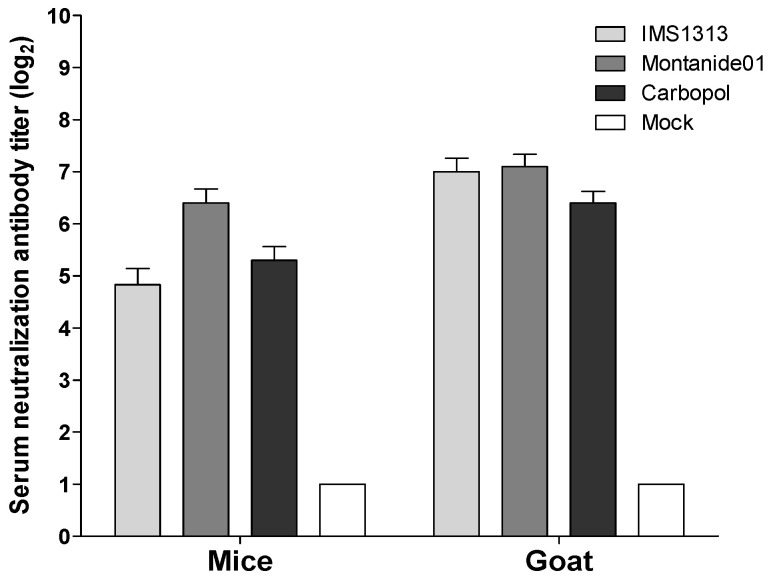
Antibody titers in mice and goats inoculated with the inactivated KBR-1 vaccine. Anti-BCoV antibody titers at 2 weeks post-injection of the inactivated KBR-1 vaccine mixed with Carbopol, Montanide01, or IMS1313.

**Table 1 viruses-14-02376-t001:** Comparison of the nucleotide and deduced amino acid sequences of the BCoV spike and hemagglutinin esterase genes according to year.

Strain	Gene	Korean BCoV Strains ^a^								
		1994	2002	2004	2015	2017	2018	2019	2020	2021
Mebus	Spike	99.5–99.6 ^b^	97.5–98.2	97.9–98.1	-	97.3–97.8	97.5–97.6	97.6–97.7	97.5–97.6	97.5–97.6
		(98.8–98.9) ^c^	(96.1–97.6)	(97.1–97.5)	-	(96.4–97.6)	(97.1–97.4)	(97.0)	(97.2–97.4)	(97.1–97.4)
	HE	99.8	97.5–98.0	97.7–98.0	97.9–98.0	97.6–98.0	97.6–97.8	96.9–97.6	97.3–98.0	97.5–97.8
		(99.3–99.5)	(97.4–98.6)	(98.6–99.1)	(98.8)	(98.1–98.8)	(98.4–98.6)	(97.1–97.9)	(97.4–98.8)	(98.1–98.6)

^a^ The annual detection lists of Korean BCoV strains are shown in Figure 1 (spike genes) and in Figure 2 (HE genes). ^b^ Nucleotide sequence homology (%). ^c^ Amino acid sequence homology (%).

**Table 2 viruses-14-02376-t002:** Comparison of the deduced amino acid sequences for the Spike genes of Korean BCoVs.

	Ref. Strain	Korean BCoV Strains ^a^								
Year	1971	1994	2002	2004	2005	2017	2018	2019	2020	2021
Spike aa	Mebus	BCoV (*n* = 2)	BCoV (*n* = 19)	BCoV (*n* = 10)	BCoV (*n* = 0)	BCoV (*n* = 13)	BCoV (*n* = 3)	BCoV (*n* = 4)	BCoV (*n* = 5)	BCoV (*n* = 6)
11	M	*	*/T	*/T	-	T	T	T	T	T
16	I	*	*	*	-	*	*	*/V	*/V	*/V
33	A	*	V	V	-	V	V	V	V	V
40	I	*/T	T	T	-	T	T	T	T	T
88	R	*	T	*/T	-	T	T	T	T	T
100	I	*	T	T	-	T	T	T	T	T
115	K	*	D	D	-	D	D	D	D	D
147	L	*	F	F	-	F	F	F	F	F
149	N	*	S	S	-	S	S	S	S	S
169	H	*	N	N	-	N	N	N	N	N
173	H	*	N	*/N	-	N	N	N	N	N
179	K	R/Q	R/Q	Q	-	Q	Q	Q	Q	Q
248	L	*	M	M	-	M	M	M	M	M
253	S	*/Y	N	N	-	N	N	N	N	N
256	M	*	*L	L	-	L	L	L	L	L
458	F	S	S	S	-	S	S	S	S	S
465	V	*	A	*/A	-	A	A	A	A	A
470	H	*	D	D	-	*/D	D	D	D	D
484	S	T	T	T	-	T	T	T	T	T
499	N	*	*/S/T/K	S/T	-	T	T	T	T	T
501	P	*	*/S	S	-	S/Y	S	S	F/S	*/S
510	S	*	T	T	-	T	T	T	T	T
531	N	D	*/D	*/D	-	D	D	D	D	D
543	S	*	A	A	-	A	A	A	A	A
578	T	*	S	S	-	S	S	S	S	S
769	A	*	S	S	-	S	S	S	S	S
779	A	N	N	N	-	N	N	N	N	N
965	V	*	E	E	-	E	E	E	E	E
984	L	W	W	W	-	W	W	W	W	W
988	V	A	A	A	-	A	A	A	A	A
1026	D	*	G	G	-	G	G	G	G	G
1030	E	*	*/D	D	-	D	D	D	D	D
1100	V	A	A	A	-	A	A	A	A	A
1241	H	*	P	*/P	-	P	P	P	P	P
1341	I	*	*/K	K	-	K	K	K	K	K

^a^ The annual detection lists of Korean BCoV strains are shown in Figure 1 (spike genes). Asterisks (*) indicate no substitution.

**Table 3 viruses-14-02376-t003:** Comparison of the deduced amino acid sequences for the HE genes of Korean BCoVs.

	Ref. Strain	Korean BCoV Strains ^a^								
Year	1971	1994	2002	2004	2005	2017	2018	2019	2020	2021
HE aa	Mebus	BCoV (*n* = 3)	BCoV (*n* = 19)	BCoV (*n* = 10)	BCoV (*n* = 2)	BCoV (*n* = 13)	BCoV (*n* = 3)	BCoV (*n* = 4)	BCoV (*n* = 5)	BCoV (*n* = 6)
5	L	*	P	*/P	P	P	P	*/P	P/H	P/H
29	N	*	T	T	T	T	T	T	T	T
46	D	*	G	G	G	G	G	G	*/G	G
339	S	*	N	N	S	N	N	*/N	*/N	*/N
367	S	*	P	P	D	P	P	P	P	P
392	L	*	I	I	I	I	I	I	I	I

^a^ The annual detection lists of Korean BCoV strains are shown in Figure 2 (HE genes). Asterisks (*) indicate no substitution.

**Table 4 viruses-14-02376-t004:** Detection of BCoV antigen and antibodies, and diarrhea scores after vaccination and challenge with BCoV KBR-1.

Group	No. Cow	Vaccination			Challenge (after Vaccination) ^a^						
		DPV 0	DPV 21	DPV 35	DPC 1	DPC 2	DPC 3	DPC 6	DPC 9	DPC 12	DPC 14
KBR-1 (10^6.0^TCID_50_/dose)	A1	^b^/<10 ^c^/0 ^d^	/40/0	/80/0	-/NT/0	-/NT/0	-/80/0	-/160/0	-/160/0	-/160/0	-/160/0
	A2	-/<10/0	/20/0	/40/0	-/NT/0	-/NT/0	-/160/0	-/80/0	-/160/0	-/160/0	-/160/0
KBR-1 (10^4.0^TCID_50_/dose)	B1	-/<20/0	/20/0	/20/0	30.3/NT/0	28.7/NT/0	33.1/40/2	-/40/1	-/80/0	-/40/0	-/80/0
	B2	-/<10/0	/10/0	/20/0	-/NT/0	29.1/NT/0	32.9/40/2	30.8/20/2	-/40/0	-/40/0	-/40/0
Mock	C1	-/<20/0	/10/0	/10/0	-/NT/0	-/NT/1	23.4/10/1	20.7/10/2	26.8/20/2	31.1/10/2	-/20/1
	C2	-/<10/0	/10/0	/10/0	-/NT/0	28.1/NT/1	22.5/20/2	23.1/10/2	29.2/20/1	-/20/2	-/40/0

^a^ Challenge (virulent BCoV: 10^6.0^ TCID_50_/mL); ^b^ Real-time RT-PCR (ct value); ^c^ HI (hemagglutinin inhibition) titer; ^d^ diarrhea score (0: normal, 1: mild, 2: moderate, 3: watery).

**Table 5 viruses-14-02376-t005:** Detection of BCoV in organs of vaccinated calves challenged with BCV KBR-1.

Group	No. Cow	Organs												
		Br *	Lv	Sp	Ln	Ht	Kd	SI	Ts	Li	Dd	Ji	Il	Ml
KBR-1 (10^6.0^TCID_50_/mL)	A1	-	-	-	-	-	-	-	-	-	-	-	-	-
	A2	-	-	-	-	-	-	-	-	-	-	-	-	-
KBR-1 (10^4.0^TCID_50_/mL)	B1	-	-	-	-	-	-	-	-	32.6	-	28.9	26.1	-
	B2	-	-	-	-	-	-	-	-	33.2	-	30.5	29.8	-
Mock	C1	-	-	-	-	-	-	-	-	-	-	23.6	26.2	-
	C2	-	-	-	-	-	-	-	-	-	31.8	24.7	25.5	-

* Br., brain; Lv., liver; Sp., spleen; Ln., lung; Ht., heart; Kd., kidney; Sl., submandibular lymph node; Ts., tonsil; Li., large intestine; Dd., duodenum; Jj., jejunum; Il., ileum; Ml., mesenteric lymph nodes.

## Data Availability

The nucleotide sequences of the 2017–2018 viruses obtained in this study were submitted to the GenBank database under accession numbers OP186313-OP186328 for the S gene and OP186329-OP186344 for the HE gene.

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
