# Peer review of "Isolation and Genetic Characterization of a Bovine Coronavirus KBR-1 Strain from Calf Feces in South Korea"

_viruses, 2022, doi:10.3390/v14112376_

Round 1

Reviewer 1 Report

Comments to the author:

Shin et al. performed molecular surveillance of BCoV in South Korea and identified that 16 calf diarrheic feces samples collected between 2017-2018 were positive for BCoV. Phylogenetic analysis of the complete S and HE genes revealed that the 16 Korean BCoV strains were genetically different from the Korean BCoV vaccine strain. The authors conducted the viral isolation of one Korean BCoV strain and then they used the inactivated strain to evaluate the immune effect. This is helpful for understanding the genetic evolution, and for preventing and controlling BCoV in South Korea. However, there are many problems in the manuscript.

Major concerns:

1.     The experimental procedures and results were not clearly described.

l  The procedure of viral isolation and culture were not provided, e.g. the concentration of trypsin used, the viral incubation time, and the details of cell state during virus culture.

l  Whether the KBR-1 isolate was purified and what was the viral purification procedure; how to determine the TCID50 of the non-cytopathic effect of the isolated strain.

l  The author should state the basis and rationality of the reference sequences selected to construct the evolutionary tree, as the type, origin and number of the BCoV sequences could bias the results. In the GenBank database, there are more than 140 BCoV genomes, more than 250 complete or nearly complete S genes (60 Korean BCoV S genes), and more than 200 complete or nearly complete HE genes (70 Korean BCoV HE genes). How do the authors ensure that the selected reference sequences were representative?

l  In the challenge experiment of calves, the selection criteria of the experimental calves were not provided, and whether the calves were negative for BCoV antigen and antibody. Moreover, the background of the challenge virulent BCoV strain was not clarified, and the detailed clinical manifestations and histopathology of the infected calves in the control group were not described.

2.     There are problems with the means of evaluating the immune effect.

l  The author should use the real-time RT-PCR method for the BCoV quantitative analysis in different organs rather than the simple qualitative analysis.

l  In the process of determining antibody titer, the neutralization experiment was only used to determine mouse serum but missed the goat serum. Besides, can the authors make sure that the description in par.2.6 “BCoV-specific antibody titers in serum were measured in IFA” is correct? IFA should be used to detect the remaining antigen after serum neutralization.

3.     The conclusions cannot be supported by the results.

l  In this study, the antigenicity and immune protection effect of KBR-1 and BC94 strain were not compared, and the results of the immunization experiment only proved that KBR-1 had a certain protective effect on the challenge virulent BCoV two weeks after the second immunization. Besides, the immune protection period was not measured, and the number of experimental animals was small. Nor has a detailed safety evaluation of the vaccine been conducted, e.g. effect on the milk production and weight gain of bovines. Therefore, the conclusion “there were differences in antigenicity between the Korean BCoV strains isolated in 1994 and after 2002, and compared with the vaccine strains, the recently isolated KBR-1 vaccine candidate is effective and safe” was not supported by sufficient experimental data.

4.     The manuscript required extensive editing.

l  Some descriptions in the manuscript need to be proofread carefully

l  The detection method of the BCoV strains and the sensitivity of the method was not described.

l  The passages of the KBR-1 strain used to select adjuvant and to evaluate vaccine efficacy should be stated. Whether the ratios of 3 adjuvants to antigens (v/v) were optimized. The authors concluded that the inactivated KBR-1 strain generated higher antibody titers when mixed with the Montanide01 (50% v/v) adjuvant than when mixed with the Carbopol (10% v/v) or IMS1313 (10% v/v); whether this result is caused by the amount of adjuvant added

l  The authors should compare the 16 Korean BCoV strains of this study with the most recently reported strains circulating in Korea (Kim et al. Prevalence and genetic characterization of bovine coronavirus identified from diarrheic pre-weaned native Korean calves from 2019 to 2021. Infect Genet Evol).

l  The authors should analyze what the reason causing 16 BCoV S and HE gene sequences to cluster into an independent clade in the discussion.

l  In par.2.5, the authors described that blood, nasal and fecal samples were collected at 0, 21, and 35 DPV, and at 1, 2, 3, 6, 9, 12, and 14 DPC to measure antigen levels, but the corresponding results were not provided.

l  The authors should analyze the probable reason why the Montanide01 (50% v/v) adjuvant is superior to Carbopol (10% v/v) or IMS1313 (10% v/v) in the discussion.

l  The discussion part should focus on discussing the experimental results obtained in this study rather than the accumulation of the literature unrelated to the experimental results.

5.     This manuscript needs to be revised by a native English speaker.

Author Response

Reviewer 1

Shin et al. performed molecular surveillance of BCoV in South Korea and identified that 16 calf diarrheic feces samples collected between 2017-2018 were positive for BCoV. Phylogenetic analysis of the complete S and HE genes revealed that the 16 Korean BCoV strains were genetically different from the Korean BCoV vaccine strain. The authors conducted the viral isolation of one Korean BCoV strain and then they used the inactivated strain to evaluate the immune effect. This is helpful for understanding the genetic evolution, and for preventing and controlling BCoV in South Korea. However, there are many problems in the manuscript.

Answer: We thank the reviewer for their helpful comments. The manuscript has been revised accordingly (see below).

Major concerns:

  1. The experimental procedures and results were not clearly described.

Comment 1-1) The procedure of viral isolation and culture were not provided, e.g. the concentration of trypsin used, the viral incubation time, and the details of cell state during virus culture.

Answer: Thank you. We have now provided details of the procedures used for virus isolation and culture (revised manuscript lines 84–109).

Comment 1-2) Whether the KBR-1 isolate was purified and what was the viral purification procedure; how to determine the TCID50 of the non-cytopathic effect of the isolated strain.

Answer: Thank you for the important question. Up until passage 4, the KBR-1 isolate was treated with antibiotics and passed through a 0.45 um filter to remove any bacteria and/or parasites (e.g., Escherichia coli, Cryptosporidium, Salmonella, etc) that may be contained in the calf diarrhea sample. At every passage, we performed PCR using a LiliFTM BD-Multi RT-PCR kit (iNtRON Biotechnology, Seongnam, Republic of Korea) to check for contamination by major viral enteric viruses (e.g., rotavirus, coronavirus, and bovine viral diarrhea virus). Virus titers were determined using the Reed and Muench method, and positive cells were counted by IFA after immunoreaction with a BCoV-specific monoclonal antibody. We have revised section 2.3 accordingly (revised manuscript lines 84–109).

Comment 1-3) The author should state the basis and rationality of the reference sequences selected to construct the evolutionary tree, as the type, origin and number of the BCoV sequences could bias the results. In the GenBank database, there are more than 140 BCoV genomes, more than 250 complete or nearly complete S genes (≥60 Korean BCoV S genes), and more than 200 complete or nearly complete HE genes (≥70 Korean BCoV HE genes). How do the authors ensure that the selected reference sequences were representative?

Answer: We selected all domestic BCoV strains (1994–2021), including the most recent strains that emerged in 2019–2021. Other reference sequences isolated in diverse regions (e.g., Asia, North America, and Europe) were based on a number of SCI papers reporting BCoV phylogenetic trees.

Comment 1-4) In the challenge experiment of calves, the selection criteria of the experimental calves were not provided, and whether the calves were negative for BCoV antigen and antibody.

Answer: All six calves used for vaccine efficacy evaluation were negative for BCoV antigen and antibodies. Before the experiment, we tested 37 calves and selected six antibody/antigen-negative animals. It is not easy to find BCoV antibody-negative calves because vaccination is widespread on Korean cattle farms. To screen the negative calves for BCoV, we used a commercial PCR kit (LiliF-BD-Multi RT-PCR kit, iNtRON Biotechnology, Seongnam, Republic of Korea) and a rapid diagnostic kit (Rapid BoviD-5 Ag, BIONOTE Inc., Hwaseong, Republic of Korea) to detect BCoV antigens in rectal and nasal swabs. The HI test was used to screen calves that were negative for serum anti-BCoV antibodies. We have described these procedures in section 2.5 (revised manuscript lines 124–128).

Comment 1-5) Moreover, the background of the challenge virulent BCoV strain was not clarified.

Answer: Thank you for pointing this out. We used the BR-1 strain as the challenge BCoV strain. Initially, the virulent BR-1 strain was isolated from the feces of a naturally-infected Korean calf in 2017, and we obtained it from the KVCC (Korea Veterinary Culture Collection, Republic of Korea, KVCC number: VR1800062). We revised the text with respect to challenge with the virulent BCoV strain, and acknowledged the source of the virus (revised manuscript lines 134–135 and 375–376).

Comment 1-6) The detailed clinical manifestations and histopathology of the infected calves in the control group were not described.

Answer: Thank you for pointing this out. We have described the clinical manifestations (diarrhea score, antigen, and antibody levels) observed in control calves (revised manuscript lines 263–271).

  1. There are problems with the means of evaluating the immune effect.

Comment 2-1) The author should use the real-time RT-PCR method for the BCoV quantitative analysis in different organs rather than the simple qualitative analysis.

Answer: Thank you for pointing this out. We have described the real-time RT-PCR in section 2.7 (revised manuscript lines 151–161).

Comment 2-2) In the process of determining antibody titer, the neutralization experiment was only used to determine mouse serum but missed the goat serum.

Answer: Thank you. Goat serum was used in addition to mouse serum. The text has been revised as follows: “Mouse and goat sera were diluted…” (revised manuscript lines 141–143).

Comment 2-3) Besides, can the authors make sure that the description in par.2.6 “BCoV-specific antibody titers in serum were measured in IFA” is correct? IFA should be used to detect the remaining antigen after serum neutralization.

Answer: Thank you for pointing this out. We have corrected the sentence (revised manuscript lines 144–147).

  1. The conclusions cannot be supported by the results.

 In this study, the antigenicity and immune protection effect of KBR-1 and BC94 strain were not compared, and the results of the immunization experiment only proved that KBR-1 had a certain protective effect on the challenge virulent BCoV two weeks after the second immunization. Besides, the immune protection period was not measured, and the number of experimental animals was small. Nor has a detailed safety evaluation of the vaccine been conducted, e.g. effect on the milk production and weight gain of bovines. Therefore, the conclusion “there were differences in antigenicity between the Korean BCoV strains isolated in 1994 and after 2002, and compared with the vaccine strains, the recently isolated KBR-1 vaccine candidate is effective and safe” was not supported by sufficient experimental data.

Answer: Thank you. We fully agree that our results are not sufficient to compare the vaccine efficacy of the KBR-1 strain with that of BC94. For clear vaccine evaluation (immunogenicity, safety, persistence, etc.) of KBR-1, further experiments will need to be conducted by the vaccine and animal medicine companies. Therefore, we have deleted all references to vaccine BC94, and suggest only that KBR-1 may be a candidate BCoV vaccine.

  1. The manuscript required extensive editing.

  Some descriptions in the manuscript need to be proofread carefully。

Comment 4-1) The detection method of the BCoV strains and the sensitivity of the method was not described.

Answer: We have revised the manuscript accordingly (lines 70–78).

Comment 4-2) The passages of the KBR-1 strain used to select adjuvant and to evaluate vaccine efficacy should be stated.

Answer: The KBR-1 strain at passage 40 was used to select the adjuvant (using mice and goats), and to evaluate vaccine efficacy in calves. The manuscript text has been revised accordingly (revised manuscript lines 108–109).

Comment 4-3) Whether the ratios of 3 adjuvants to antigens (v/v) were optimized. The authors concluded that the inactivated KBR-1 strain generated higher antibody titers when mixed with the Montanide01 (50% v/v) adjuvant than when mixed with the Carbopol (10% v/v) or IMS1313 (10% v/v); whether this result is caused by the amount of adjuvant added.

Answer: We used three adjuvants at the following ratio: Montanide01 (10% v/v), IMS1313 (30% v/v), and Carbopol (0.02% w/v). The amount of each adjuvant was based on the manufacturer’s manual (SEPPIC, France) and a related reference (Dey et al., 2012, Vaccine). The final titer of the inactivated KBR-1 was optimized at 105.0 TCID50/dose. We have revised the manuscript text accordingly and included an additional reference (revised manuscript lines 113–116).

Comment 4-4) The authors should compare the 16 Korean BCoV strains of this study with the most recently reported strains circulating in Korea (Kim et al. Prevalence and genetic characterization of bovine coronavirus identified from diarrheic pre-weaned native Korean calves from 2019 to 2021. Infect Genet Evol).

Answer: We agree with reviewer’s opinion that the recently reported Korean BCoV strains should be included in our study. Therefore, we have included these recent Korean BCoV strains (2019–2021) and a revised phylogenetic tree for the spike and HE genes (see the revised Figures 1 and 2).

Comment 4-5) The authors should analyze what the reason causing 16 BCoV S and HE gene sequences to cluster into an independent clade in the discussion.

Answer: We have revised the Discussion section accordingly (lines 312–318).

Comment 4-6) In par.2.5, the authors described that blood, nasal and fecal samples were collected at 0, 21, and 35 DPV, and at 1, 2, 3, 6, 9, 12, and 14 DPC to measure antigen levels, but the corresponding results were not provided.

Answer: Real-time PCR scores have been added to part 3.6, mentioned in the Discussion section, and included in Tables 4 and 5 (revised manuscript lines 267–271, 282–286, and 344–353).

Comment 4-7) The authors should analyze the probable reason why the Montanide01 (50% v/v) adjuvant is superior to Carbopol (10% v/v) or IMS1313 (10% v/v) in the discussion.

Answer: We have mentioned this in the Discussion section (lines 329–339).

Comment 4-8) The discussion part should focus on discussing the experimental results obtained in this study rather than the accumulation of the literature unrelated to the experimental results.

Answer: We have revised the Discussion section accordingly (lines 298-299, 301–309, 312–318, 329–339, and 344–353).

  1. This manuscript needs to be revised by a native English speaker.

Answer: The revised manuscript has been revised by a commercial English editing company (www.bioedit.com).

Reviewer 2 Report

Authors have done good work but  following queries need to be solved first before publication:

1) How many times authors have done the sequencing with howmany sample size? (howmany replicates and howmany repeatation?) Sanger sequencing by PCR products is not reliable to identified mutation reportation an complete coverage of gene, however , it is acceptable if authors have done extensive multiple time sequencing and getting same result! if not done, authors are requested to do it.

2) Authors need to understand that genetic determinant can not be always the antigenic determinant/ Immunogenic determinant. phylogenetic distict group can not be concluded as there is need of different vaccine or previous vaccine can't give protection or earlier isolate's based vaccine is inefficient in protection or any comment/staement regarding the vaccine efficacy until and unless proper experiment ,

Did authors have done heterologous serum/virus neutralization test?

Did authors have done heterologous HI  testing or Cross protective virus challenge study ?

If I have missed in manuscript, please let me know, but if not done, manuscript need to completely revised and removed the any word/phrase/statement regarding previous vaccine and current out break which can't give protection or as authors understanding,

The conclusion can be made in a way that   "need of new vaccine", "need vaccine based on current circulating strain", update in vaccine as per the current prevalence or anything! without any statment which give impression that previous vaccine do not give protection (because it is not yet studied by your group)

3) why  human rectal tumor (HRT-18) cell line used , there are many though !

4) Authors did not mention the data/ graph of viral shedding copy number in comparision of both dose vaccine with the control groups.

5) Authors did not conducted IHC or Histopathology of all groups? why ? if possible, authors are requested to do so.

Author Response

Reviewer 2

Authors have done good work but following queries need to be solved first before publication:

Comment 1) How many times authors have done the sequencing with how many sample size? (how many replicates and how many repeatation?) Sanger sequencing by PCR products is not reliable to identified mutation reportation an complete coverage of gene, however, it is acceptable if authors have done extensive multiple time sequencing and getting same result! if not done, authors are requested to do it.

Answer: We agree with the reviewer’s opinion and recognize that sequencing should be repeated multiple times. Therefore, we collected 40–60 mg of feces from each calf and sequenced each cDNA 3–4 times using sequencing primers and gene-specific primers. We have described this in the revised manuscript (lines 67-68 and 74–78).

Comment 2) Authors need to understand that genetic determinant can not be always the antigenic determinant/ Immunogenic determinant. phylogenetic distict group can not be concluded as there is need of different vaccine or previous vaccine can't give protection or earlier isolate's based vaccine is inefficient in protection or any comment/staement regarding the vaccine efficacy until and unless proper experiment,

Did authors have done heterologous serum/virus neutralization test?

Did authors have done heterologous HI testing or Cross protective virus challenge study?

If I have missed in manuscript, please let me know, but if not done, manuscript need to completely revised and removed the any word/phrase/statement regarding previous vaccine and current out break which can't give protection or as authors understanding,

The conclusion can be made in a way that   "need of new vaccine", "need vaccine based on current circulating strain", update in vaccine as per the current prevalence or anything! without any statment which give impression that previous vaccine do not give protection (because it is not yet studied by your group)

Answer: Thank you. We agree that our results are not strong enough to suggest the need to replace the current vaccine with KBR-1. We did not perform a heterologous serum/virus neutralization test, heterologous HI testing, or cross-protective virus challenge studies. Therefore, we have deleted all reference to the previous vaccine (BC94), and now suggest only that KBR-1 is a potential candidate BCoV vaccine.

Comment 3) Why human rectal tumor (HRT-18) cell line used, there are many though!

Answer: BCoV has been isolated and propagated in several cell lines, including Vero (African green monkey kidney), MDBK (Madin-Darby bovine kidney), HmLu-1 (Hamster lung), and HRT-18 (human rectal tumor). HRT-18 is the most sensitive cell used for primary isolation, mainly in the presence of trypsin. We also tried isolating BCoV using other cells (MDBK and Vero). Finally, we isolated KBR-1 after 3–4 blind passages in HRT-18 cells.

Comment 4) Authors did not mention the data/ graph of viral shedding copy number in comparision of both dose vaccine with the control groups.

Answer: The real-time PCR scores have been included in section 3.6, mentioned in the Discussion section, and added to Tables 4 and 5 (revised manuscript lines 267–271, 282–286, and 344–353).

Comment 5) Authors did not conducted IHC or Histopathology of all groups? Why? if possible, authors are requested to do so.

Answer: Unfortunately, we did not perform IHC and histopathological experiments in this study. We intend to do so in future studies.

Round 2

Reviewer 1 Report

Accept in present form.

Author Response

FW: [Viruses] Manuscript ID: viruses-1921401 - Accept with Minor Revisions

Additional comments are described as below.

  1. Introduction

1-1) P1, L31-42: The author should summarize broad information about virological characteristics of BCoV.

RESPONSE: Thank you for pointing this out. As requested, we have summarized the characteristics of BCoV (lines 31–32).

1-2) P1, L42: The author should split up the paragraph at this point.

RESPONSE: As requested, we have split the paragraph from L42 (line 41).

1-3) P12, L54: The author should provide the information about BCoV vaccine and the rationale for the need to develop new inactivated vaccine in a new paragraph.

RESPONSE: Thank you. We have added the following sentence to the revised manuscript (lines 52–54):

  1. Materials and Methods

2-1) P2, L82: The author should state the basis and rationality of the reference sequences selected to construct the phylogenetic tree.

RESPONSE: We have added the following sentence to the revised manuscript (lines 86–88).

  1. Results

3-1) P8. L226-228: The basis of KBR-1 strain selection as vaccine candidate was not described.

RESPONSE: Thank you for pointing this out. We attempted to isolate BCoV from 16 BCoV-positive diarrhea samples, but unfortunately only the KBR-1 strain could be isolated and cultured continuously in vitro.

  1. Discussion

4-1) P10, L290: The author should discuss prevalence of BCoV and compare the results with other studies in a new paragraph. The necessity and importance of study on BCoV should be described.

RESPONSE: We agree with the reviewer’s comment that, based on the prevalence of BCoV in Korea, we need to emphasize the necessity and importance of BCoV studies, including those in other countries. We have revised the Discussion section accordingly (lines 296–305).

4-2) P10, L291-298: This part seems duplicated contents and the description of the literature unrelated to the experimental results.

RESPONSE: We have deleted these sentences from the original manuscript (lines 291–298).

4-3) P11, L319-329: In this study, the diarrhea samples were obtained from calves. Therefore, the literature about the vaccination on calf should be described. Please remove the sentence about unrelated literature on this part. Moreover, the author should focus on discussing the experimental results obtained in this study.

RESPONSE: As requested, we have removed the sentence describing unrelated literature, and have referred to literature describing vaccination of calves (lines 327–335, and 358–359).

4-4) P11, L329: The author should split up the paragraph at this point.

RESPONSE: As requested, we have we split the paragraph (line 327).

Reviewer 2 Report

Authors have answered all question, However IHC was not done in experimental challenge study as well as immunological parameters of host were not studied after challenge. Therefore, Title doesnt't require Immunological, Please remove 'and immunological ' from the title. 

so it will be " Isolation and genetic characterization of a bovine coronavirus KBR-1 strain from calf feces in South Korea"

Or

Please justify it for ...'as it is' consideration.

valid reason will be consider.

Author Response

Reviewer 2

Comment 1: Authors have answered all question, However IHC was not done in experimental challenge study as well as immunological parameters of host were not studied after challenge. Therefore, Title doesnt't require Immunological, Please remove 'and immunological ' from the title. 

so it will be " Isolation and genetic characterization of a bovine coronavirus KBR-1 strain from calf feces in South Korea"

Or

Please justify it for ...'as it is' consideration.

valid reason will be consider.

Answer: According to reviewer’s comment, we changed to the title “Isolation and genetic characterization of a bovine coronavirus KBR-1 strain from calf feces in South Korea”.
